# Towards Accurate Binary Convolutional Neural Network

**Xiaofan Lin**        **Cong Zhao**        **Wei Pan***
DJI Innovations Inc, Shenzhen, China
{xiaofan.lin, cong.zhao, wei.pan}@dji.com

## Abstract

We introduce a novel scheme to train binary convolutional neural networks (CNNs) – CNNs with weights and activations constrained to {-1,+1} at run-time. It has been known that using binary weights and activations drastically reduce memory size and accesses, and can replace arithmetic operations with more efficient bitwise operations, leading to much faster test-time inference and lower power consumption. However, previous works on binarizing CNNs usually result in severe prediction accuracy degradation. In this paper, we address this issue with two major innovations: (1) approximating full-precision weights with the linear combination of multiple binary weight bases; (2) employing multiple binary activations to alleviate information loss. The implementation of the resulting binary CNN, denoted as ABC-Net, is shown to achieve much closer performance to its full-precision counterpart, and even reach the comparable prediction accuracy on ImageNet and forest trail datasets, given adequate binary weight bases and activations.

## 1   Introduction

Convolutional neural networks (CNNs) have achieved state-of-the-art results on real-world applications such as image classification [He et al., 2016] and object detection [Ren et al., 2015], with the best results obtained with large models and sufficient computation resources. Concurrent to these progresses, the deployment of CNNs on mobile devices for consumer applications is gaining more and more attention, due to the widespread commercial value and the exciting prospect.

On mobile applications, it is typically assumed that training is performed on the server and test or inference is executed on the mobile devices [Courbariaux et al., 2016, Esser et al., 2016]. In the training phase, GPUs enabled substantial breakthroughs because of their greater computational speed. In the test phase, however, GPUs are usually too expensive to deploy. Thus improving the test-time performance and reducing hardware costs are likely to be crucial for further progress, as mobile applications usually require real-time, low power consumption and fully embeddable. As a result, there is much interest in research and development of dedicated hardware for deep neural networks (DNNs). Binary neural networks (BNNs) [Courbariaux et al., 2016, Rastegari et al., 2016], i.e., neural networks with weights and perhaps activations constrained to only two possible values (e.g., -1 or +1), would bring great benefits to specialized DNN hardware for three major reasons: (1) the binary weights/activations reduce memory usage and model size 32 times compared to single-precision version; (2) if weights are binary, then most multiply-accumulate operations can be replaced by simple accumulations, which is beneficial because multipliers are the most space and power-hungry components of the digital implementation of neural networks; (3) furthermore, if both activations and weights are binary, the multiply-accumulations can be replaced by the bitwise operations: xnor and bitcount Courbariaux et al. [2016]. This could have a big impact on dedicated deep learning hardware. For instance, a 32-bit floating point multiplier costs about 200 Xilinx FPGA slices [Govindu et al., 2004], whereas a 1-bit xnor gate only costs a single slice. Semiconductor

manufacturers like IBM [Esser et al., 2016] and Intel [Venkatesh et al., 2016] have been involved in the research and development of related chips.

However, binarization usually cause severe prediction accuracy degradation, especially on complex tasks such as classification on ImageNet dataset. To take a closer look, Rastegari et al. [2016] shows that binarizing weights causes the accuracy of Resnet-18 drops from 69.3% to 60.8% on ImageNet dataset. If further binarize activations, the accuracy drops to 51.2%. Similar phenomenon can also be found in literatures such as [Hubara et al., 2016]. Clearly there is a considerable gap between the accuracy of a full-precision model and a binary model.

This paper proposes a novel scheme for binarizing CNNs, which aims to alleviate, or even eliminate the accuracy degradation, while still significantly reducing inference time, resource requirement and power consumption. The paper makes the following major contributions.

- We approximate full-precision weights with the linear combination of multiple binary weight bases. The weights values of CNNs are constrained to $\{-1, +1\}$, which means convolutions can be implemented by only addition and subtraction (without multiplication), or bitwise operation when activations are binary as well. We demonstrate that 3~5 binary weight bases are adequate to well approximate the full-precision weights.

- We introduce multiple binary activations. Previous works have shown that the quantization of activations, especially binarization, is more difficult than that of weights [Cai et al., 2017, Courbariaux et al., 2016]. By employing five binary activations, we have been able to reduce the Top-1 and Top-5 accuracy degradation caused by binarization to around 5% on ImageNet compared to the full precision counterpart.

It is worth noting that the multiple binary weight bases/activations scheme is preferable to the fixed-point quantization in previous works. In those fixed-point quantized networks one still needs to employ arithmetic operations, such as multiplication and addition, on fixed-point values. Even though faster than floating point, they still require relatively complex logic and can consume a lot of power. Detailed discussions can be found in Section 5.2.

Ideally, combining more binary weight bases and activations always leads to better accuracy and will eventually get very close to that of full-precision networks. We verify this on ImageNet using Resnet network topology. **This is the first time a binary neural network achieves prediction accuracy comparable to its full-precision counterpart on ImageNet.**

## 2 Related work

**Quantized Neural Networks:** High precision parameters are not very necessary to reach high performance in deep neural networks. Recent research efforts (e.g., [Hubara et al., 2016]) have considerably reduced a large amounts of memory requirement and computation complexity by using low bitwidth weights and activations. Zhou et al. [2016] further generalized these schemes and proposed to train CNNs with low bitwidth gradients. By performing the quantization after network training or using the "straight-through estimator (STE)" [Bengio et al., 2013], these works avoided the issues of non-differentiable optimization. While some of these methods have produced good results on datasets such as CIFAR-10 and SVHN, none has produced low precision networks competitive with full-precision models on large-scale classification tasks, such as ImageNet. In fact, [Zhou et al., 2016] and [Hubara et al., 2016] experiment with different combinations of bitwidth for weights and activations, and show that the performance of their highly quantized networks deteriorates rapidly when the weights and activations are quantized to less than 4-bit numbers. Cai et al. [2017] enhance the performance of a low bitwidth model by addressing the gradient mismatch problem, nevertheless there is still much room for improvement.

**Binarized Neural Networks:** The binary representation for deep models is not a new topic. At the emergence of artificial neural networks, inspired biologically, the unit step function has been used as the activation function [Toms, 1990]. It is known that binary activation can use spiking response for event-based computation and communication (consuming energy only when necessary) and therefore is energy-efficient [Esser et al., 2016]. Recently, Courbariaux et al. [2016] introduce Binarized-Neural-Networks (BNNs), neural networks with binary weights and activations at run-time. Different from their work, Rastegari et al. [2016] introduce simple, efficient, and accurate approximations to CNNs by binarizing the weights and even the intermediate representations in CNNs. All these works drastically reduce memory consumption, and replace most arithmetic operations with bitwise operations, which potentially lead to a substantial increase in power efficiency.

In all above mentioned works, binarization significantly reduces accuracy. Our experimental results on ImageNet show that we are close to filling the gap between the accuracy of a binary model and its full-precision counterpart. We relied on the idea of finding the best approximation of full-precision convolution using multiple binary operations, and employing multiple binary activations to allow more information passing through.

## 3 Binarization methods

In this section, we detail our binarization method, which is termed ABC-Net (**A**ccurate-**B**inary-**C**onvolutional) for convenience. Bear in mind that during training, the real-valued weights are reserved and updated at every epoch, while in test-time only binary weights are used in convolution.

### 3.1 Weight approximation

Consider a $L$-layer CNN architecture. Without loss of generality, we assume the weights of each convolutional layer are tensors of dimension $(w, h, c_{\text{in}}, c_{\text{out}})$, which represents filter width, filter height, input-channel and output-channel respectively. We propose two variations of binarization method for weights at each layer: 1) approximate weights as a whole and 2) approximate weights channel-wise.

#### 3.1.1 Approximate weights as a whole

At each layer, in order to constrain a CNN to have binary weights, we estimate the real-value weight filter $\boldsymbol{W} \in \mathbb{R}^{w \times h \times c_{\text{in}} \times c_{\text{out}}}$ using the linear combination of $M$ binary filters $\boldsymbol{B}_1, \boldsymbol{B}_2, \cdots, \boldsymbol{B}_M \in \{-1, +1\}^{w \times h \times c_{\text{in}} \times c_{\text{out}}}$ such that $\boldsymbol{W} \approx \alpha_1 \boldsymbol{B}_1 + \alpha_2 \boldsymbol{B}_2 + \cdots + \alpha_M \boldsymbol{B}_M$. To find an optimal estimation, a straightforward way is to solve the following optimization problem:

$$\min_{\boldsymbol{\alpha}, \boldsymbol{B}} J(\boldsymbol{\alpha}, \boldsymbol{B}) = ||\boldsymbol{w} - \boldsymbol{B}\boldsymbol{\alpha}||^2, \quad \text{s.t. } \boldsymbol{B}_{ij} \in \{-1, +1\}, \tag{1}$$

where $\boldsymbol{B} = [\text{vec}(\boldsymbol{B}_1), \text{vec}(\boldsymbol{B}_2), \cdots, \text{vec}(\boldsymbol{B}_M)]$, $\boldsymbol{w} = \text{vec}(\boldsymbol{W})$ and $\boldsymbol{\alpha} = [\alpha_1, \alpha_2, \cdots, \alpha_M]^{\text{T}}$. Here the notation $\text{vec}(\cdot)$ refers to vectorization.

Although a local minimum solution to (1) can be obtained by numerical methods, one could not backpropagate through it to update the real-value weight filter $\boldsymbol{W}$. To address this issue, assuming the mean and standard deviation of $\boldsymbol{W}$ are $\text{mean}(\boldsymbol{W})$ and $\text{std}(\boldsymbol{W})$ respectively, we fix $\boldsymbol{B}_i$'s as follows:

$$\boldsymbol{B}_i = F_{u_i}(\boldsymbol{W}) := \text{sign}(\bar{\boldsymbol{W}} + u_i \text{std}(\boldsymbol{W})), i = 1, 2, \cdots, M, \tag{2}$$

where $\bar{\boldsymbol{W}} = \boldsymbol{W} - \text{mean}(\boldsymbol{W})$, and $u_i$ is a shift parameter. For example, one can choose $u_i$'s to be $u_i = -1 + (i-1)\frac{2}{M-1}, i = 1, 2, \cdots, M$, to shift evenly over the range $[-\text{std}(\boldsymbol{W}), \text{std}(\boldsymbol{W})]$, or leave it to be trained by the network. This is based on the observation that the full-precision weights tend to have a symmetric, non-sparse distribution, which is close to Gaussian. To gain more intuition and illustrate the approximation effectiveness, an example is visualized in Section S2 of the supplementary material.

With $\boldsymbol{B}_i$'s chosen, (1) becomes a linear regression problem

$$\min_{\boldsymbol{\alpha}} J(\boldsymbol{\alpha}) = ||\boldsymbol{w} - \boldsymbol{B}\boldsymbol{\alpha}||^2, \tag{3}$$

in which $\boldsymbol{B}_i$'s serve as the bases in the design/dictionary matrix. We can then back-propagate through $\boldsymbol{B}_i$'s using the "straight-through estimator" (STE) [Bengio et al., 2013]. Assume $c$ as the cost function, $\boldsymbol{A}$ and $\boldsymbol{O}$ as the input and output tensor of a convolution respectively, the forward and backward approach of an approximated convolution during training can be computed as follows:

Forward: $\boldsymbol{B}_1, \boldsymbol{B}_2, \cdots, \boldsymbol{B}_M = F_{u_1}(\boldsymbol{W}), F_{u_2}(\boldsymbol{W}), \cdots, F_{u_M}(\boldsymbol{W}),$ \hfill (4)

Solve (3) for $\boldsymbol{\alpha}$, \hfill (5)

$$\boldsymbol{O} = \sum_{m=1}^{M} \alpha_m \text{Conv}(\boldsymbol{B}_m, \boldsymbol{A}). \tag{6}$$

Backward: $\dfrac{\partial c}{\partial \boldsymbol{W}} = \dfrac{\partial c}{\partial \boldsymbol{O}} \left( \sum_{m=1}^{M} \alpha_m \dfrac{\partial \boldsymbol{O}}{\partial \boldsymbol{B}_m} \dfrac{\partial \boldsymbol{B}_m}{\partial \boldsymbol{W}} \right) \overset{\text{STE}}{=} \dfrac{\partial c}{\partial \boldsymbol{O}} \left( \sum_{m=1}^{M} \alpha_m \dfrac{\partial \boldsymbol{O}}{\partial \boldsymbol{B}_m} \right) = \sum_{m=1}^{M} \alpha_m \dfrac{\partial c}{\partial \boldsymbol{B}_m}.$

\hfill (7)

In test-time, only (6) is required. The block structure of this approximated convolution layer is shown on the left side in Figure 1. With suitable hardwares and appropriate implementations, the convolution can be efficiently computed. For example, since the weight values are binary, we can implement the convolution with additions and subtractions (thus without multiplications). Furthermore, if the input $\boldsymbol{A}$ is binary as well, we can implement the convolution with bitwise operations: xnor and bitcount [Rastegari et al., 2016]. Note that the convolution with each binary filter can be computed in parallel.

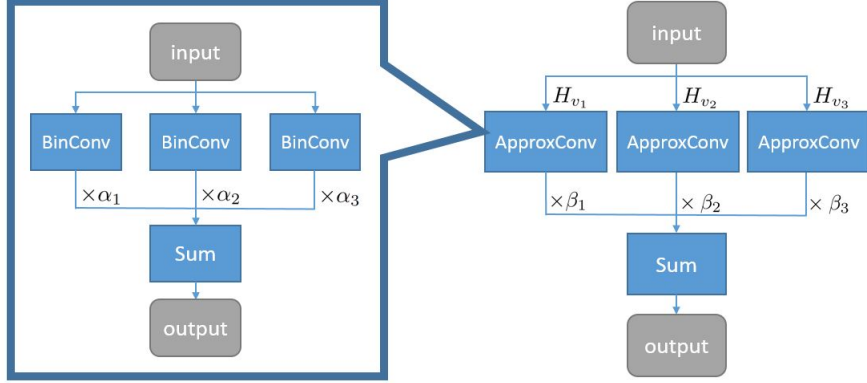

Figure 1: An example of the block structure of the convolution in ABC-Net. $M = N = 3$. On the left is the structure of the approximated convolution (ApproxConv). ApproxConv is expected to approximate the conventional full-precision convolution with linear combination of binary convolutions (BinConv), i.e., convolution with binary and weights. On the right is the overall block structure of the convolution in ABC-Net. The input is binarized using different functions $H_{v_1}, H_{v_2}, H_{v_3}$, passed into the corresponding ApproxConv's and then summed up after multiplying their corresponding $\beta_n$'s. With the input binarized, the BinConv's can be implemented with highly efficient bitwise operations. There are 9 BinConv's in this example and they can work in parallel.

### 3.1.2 Approximate weights channel-wise

Alternatively, we can estimate the real-value weight filter $\boldsymbol{W}_i \in \mathbb{R}^{w \times h \times c_{\text{in}}}$ of each output channel $i \in \{1, 2, \cdots, c_{\text{out}}\}$ using the linear combination of $M$ binary filters $\boldsymbol{B}_{i1}, \boldsymbol{B}_{i2}, \cdots, \boldsymbol{B}_{iM} \in \{-1, +1\}^{w \times h \times c_{\text{in}}}$ such that $\boldsymbol{W}_i \approx \alpha_{i1}\boldsymbol{B}_{i1} + \alpha_{i2}\boldsymbol{B}_{i2} + \cdots + \alpha_{iM}\boldsymbol{B}_{iM}$. Again, to find an optimal estimation, we solve a linear regression problem analogy to (3) for each output channel. After convolution, the results are concatenated together along the output-channel dimension. If $M = 1$, this approach reduces to the Binary-Weights-Networks (BWN) proposed in [Rastegari et al., 2016].

Compared to weights approximation as a whole, the channel-wise approach approximates weights more elaborately, however no extra cost is needed during inference. Since this approach requires more computational resources during training, we leave it as a future work and focus on the former approximation approach in this paper.

### 3.2 Multiple binary activations and bitwise convolution

As mentioned above, a convolution can be implemented without multiplications when weights are binarized. However, to utilize the bitwise operation, the activations must be binarized as well, as they are the inputs of convolutions.

Similar to the activation binarization procedure in [Zhou et al., 2016], we binarize activations after passing it through a bounded activation function $h$, which ensures $h(x) \in [0, 1]$. We choose the bounded rectifier as $h$. Formally, it can be defined as:

$$h_v(x) = \text{clip}(x + v, 0, 1), \tag{8}$$

where $v$ is a shift parameter. If $v = 0$, then $h_v$ is the clip activation function in [Zhou et al., 2016].

We constrain the binary activations to either 1 or -1. In order to transform the real-valued activation $\boldsymbol{R}$ into binary activation, we use the following binarization function:

$$H_v(\boldsymbol{R}) := 2\mathbb{I}_{h_v(\boldsymbol{R}) \geq 0.5} - 1, \tag{9}$$

where $\mathbb{I}$ is the indicator function. The conventional forward and backward approach of the activation can be given as follows:

$$\begin{aligned} \text{Forward: } & \boldsymbol{A} = H_v(\boldsymbol{R}). \\ \text{Backward: } & \frac{\partial c}{\partial \boldsymbol{R}} = \frac{\partial c}{\partial \boldsymbol{A}} \circ \mathbb{I}_{0 \le \boldsymbol{R} - v \le 1}. \quad \text{(using STE)} \end{aligned} \quad (10)$$

Here $\circ$ denotes the Hadamard product. As can be expected, binaizing activations as above is kind of crude and leads to non-trivial losses in accuracy, as shown in Rastegari et al. [2016], Hubara et al. [2016]. While it is also possible to approximate activations with linear regression, as that of weights, another critical challenge arises – unlike weights, the activations always vary in test-time inference. Luckily, this difficulty can be avoided by exploiting the statistical structure of the activations of deep networks.

Our scheme can be described as follows. First of all, to keep the distribution of activations relatively stable, we resort to batch normalization [Ioffe and Szegedy, 2015]. This is a widely used normalization technique, which forces the responses of each network layer to have zero mean and unit variance. We apply this normalization before activation. Secondly, we estimate the real-value activation $\boldsymbol{R}$ using the linear combination of $N$ binary activations $\boldsymbol{A}_1, \boldsymbol{A}_2, \cdots, \boldsymbol{A}_N$ such that $\boldsymbol{R} \approx \beta_1 \boldsymbol{A}_1 + \beta_2 \boldsymbol{A}_2 + \cdots + \beta_N \boldsymbol{A}_N$, where

$$\boldsymbol{A}_1, \boldsymbol{A}_2, \cdots, \boldsymbol{A}_N = H_{v_1}(\boldsymbol{R}), H_{v_2}(\boldsymbol{R}), \cdots, H_{v_N}(\boldsymbol{R}). \quad (11)$$

Different from that of weights, the parameters $\beta_n$'s and $v_n$'s ($n = 1, \cdots, N$) here are both trainable, just like the scale and shift parameters in batch normalization. Without the explicit linear regression approach, $\beta_n$'s and $v_n$'s are tuned by the network itself during training and fixed in test-time. They are expected to learn and utilize the statistical features of full-precision activations.

The resulting network architecture outputs multiple binary activations $\boldsymbol{A}_1, \boldsymbol{A}_2, \cdots, \boldsymbol{A}_N$ and their corresponding coefficients $\beta_1, \beta_2, \cdots, \beta_N$, which allows more information passing through compared to the former one. Combining with the weight approximation, the whole convolution scheme is given by:

$$\text{Conv}(\boldsymbol{W}, \boldsymbol{R}) \approx \text{Conv}\left(\sum_{m=1}^{M} \alpha_m \boldsymbol{B}_m, \sum_{n=1}^{N} \beta_n \boldsymbol{A}_n\right) = \sum_{m=1}^{M} \sum_{n=1}^{N} \alpha_m \beta_n \text{Conv}\left(\boldsymbol{B}_m, \boldsymbol{A}_n\right), \quad (12)$$

which suggests that it can be implemented by computing $M \times N$ bitwise convolutions in parallel. An example of the whole convolution scheme is shown in Figure 1.

### 3.3 Training algorithm

A typical block in CNN contains several different layers, which are usually in the following order: (1) Convolution, (2) Batch Normalization, (3) Activation and (4) Pooling. The batch normalization layer [Ioffe and Szegedy, 2015] normalizes the input batch by its mean and variance. The activation is an element-wise non-linear function (e.g., Sigmoid, ReLU). The pooling layer applies any type of pooling (e.g., max,min or average) on the input batch. In our experiment, we observe that applying max-pooling on binary input returns a tensor that most of its elements are equal to +1, resulting in a noticeable drop in accuracy. Similar phenomenon has been reported in Rastegari et al. [2016] as well. Therefore, we put the max-pooling layer before the batch normalization and activation.

Since our binarization scheme approximates full-precision weights, using the full-precision pre-train model serves as a perfect initialization. However, fine-tuning is always required for the weights to adapt to the new network structure. The training procedure, i.e., ABC-Net, is summarized in Section S1 of the supplementary material.

It is worth noting that as $M$ increases, the shift parameters get closer and the bases of the linear combination are more correlated, which sometimes lead to rank deficiency when solving (3). This can be tackled with the $\ell_2$ regularization.

## 4 Experiment results

In this section, the proposed ABC-Net was evaluated on the ILSVRC12 ImageNet classification dataset [Deng et al., 2009], and visual perception of forest trails datasets for mobile robots [Giusti et al., 2016] in Section S6 of supplementary material.

## 4.1 Experiment results on ImageNet dataset

The ImageNet dataset contains about 1.2 million high-resolution natural images for training that spans 1000 categories of objects. The validation set contains 50k images. We use Resnet ([He et al., 2016]) as network topology. The images are resized to 224x224 before fed into the network. We report our classification performance using Top-1 and Top-5 accuracies.

### 4.1.1 Effect of weight approximation

We first evaluate the weight approximation technique by examining the accuracy improvement for a binary model. To eliminate variables, we leave the activations being full-precision in this experiment. Table 1 shows the prediction accuracy of ABC-Net on ImageNet with different choices of $M$. For comparison, we add the results of Binary-Weights-Network (denoted 'BWN') reported in Rastegari et al. [2016] and the full-precision network (denoted 'FP'). The BWN binarizes weights and leaves the activations being full-precision as we do. All results in this experiment use Resnet-18 as network topology. It can be observed that as $M$ increases, the accuracy of ABC-Net converges to its full-precision counterpart. The Top-1 gap between them reduces to only 0.9 percentage point when $M = 5$, which suggests that this approach nearly eliminates the accuracy degradation caused by binarizing weights.

Table 1: Top-1 (left) and Top-5 (right) accuracy of ABC-Net on ImageNet, using full-precision activation and different choices of the number of binary weight bases $M$.

|       | BWN   | $M = 1$ | $M = 2$ | $M = 3$ | $M = 5$ | FP    |
|-------|-------|---------|---------|---------|---------|-------|
| Top-1 | 60.8% | 62.8%   | 63.7%   | 66.2%   | 68.3%   | 69.3% |
| Top-5 | 83.0% | 84.4%   | 85.2%   | 86.7%   | 87.9%   | 89.2% |

For interested readers, Figure S4 in section S5 of the supplementary material shows that the relationship between accuracy and $M$ appears to be linear. Also, in Section S2 of the supplementary material, a visualization of the approximated weights is provided.

### 4.1.2 Configuration space exploration

We explore the configuration space of combinations of number of weight bases and activations. Table 2 presents the results of ABC-Net with different configurations. The parameter settings for these experiments are provided in Section S4 of the supplementary material.

Table 2: Prediction accuracy (Top-1/Top-5) for ImageNet with different choices of $M$ and $N$ in a ABC-Net (approximate weights as a whole). "res18", "res34" and "res50" are short for Resnet-18, Resnet-34 and Resnet-50 network topology respectively. $M$ and $N$ refer to the number of weight bases and activations respectively.

| Network | $M$-weight base | $N$-activation base | Top-1 | Top-5 | Top-1 gap | Top-5 gap |
|---------|-----------------|---------------------|-------|-------|-----------|-----------|
| res18   | 1               | 1                   | 42.7% | 67.6% | 26.6%     | 21.6%     |
| res18   | 3               | 1                   | 49.1% | 73.8% | 20.2%     | 15.4%     |
| res18   | 3               | 3                   | 61.0% | 83.2% | 8.3%      | 6.0%      |
| res18   | 3               | 5                   | 63.1% | 84.8% | 6.2%      | 4.4%      |
| res18   | 5               | 1                   | 54.1% | 78.1% | 15.2%     | 11.1%     |
| res18   | 5               | 3                   | 62.5% | 84.2% | 6.8%      | 5.0%      |
| res18   | 5               | 5                   | 65.0% | 85.9% | **4.3%**  | **3.3%**  |
| res18   | Full Precision  |                     | 69.3% | 89.2% | -         | -         |
| res34   | 1               | 1                   | 52.4% | 76.5% | 20.9%     | 14.8%     |
| res34   | 3               | 3                   | 66.7% | 87.4% | 6.6%      | 3.9%      |
| res34   | 5               | 5                   | 68.4% | 88.2% | **4.9%**  | **3.1%**  |
| res34   | Full Precision  |                     | 73.3% | 91.3% | -         | -         |
| res50   | 5               | 5                   | 70.1% | 89.7% | **6.0%**  | **3.1%**  |
| res50   | Full Precision  |                     | 76.1% | 92.8% | -         | -         |

As balancing between multiple factors like training time and inference time, model size and accuracy is more a problem of practical trade-off, there will be no definite conclusion as which combination of

$(M, N)$ one should choose. In general, Table 2 shows that (1) the prediction accuracy of ABC-Net improves greatly as the number of binary activations increases, which is analogous to the weight approximation approach; (2) larger $M$ or $N$ gives better accuracy; (3) when $M = N = 5$, the Top-1 gap between the accuracy of a full-precision model and a binary one reduces to around 5%. To gain a visual understanding and show the possibility of extensions to other tasks such object detection, we print the a sample of feature maps in Section S3 of supplementary material.

### 4.1.3 Comparison with the state-of-the-art

Table 3: Classification test accuracy of CNNs trained on ImageNet with Resnet-18 network topology. 'W' and 'A' refer to the weight and activation bitwidth respectively.

| Model | W | A | Top-1 | Top-5 |
|---|---|---|---|---|
| Full-Precision Resnet-18 [full-precision weights and activation] | 32 | 32 | 69.3% | 89.2% |
| BWN [full-precision activation] Rastegari et al. [2016] | 1 | 32 | 60.8% | 83.0% |
| DoReFa-Net [1-bit weight and 4-bit activation] Zhou et al. [2016] | 1 | 4 | 59.2% | 81.5% |
| XNOR-Net [binary weight and activation] Rastegari et al. [2016] | 1 | 1 | 51.2% | 73.2% |
| BNN [binary weight and activation] Courbariaux et al. [2016] | 1 | 1 | 42.2% | 67.1% |
| ABC-Net [5 binary weight bases, 5 binary activations] | 1 | 1 | 65.0% | 85.9% |
| ABC-Net [5 binary weight bases, full-precision activations] | 1 | 32 | 68.3% | 87.9% |

Table 3 presents a comparison between ABC-Net and several other state-of-the-art models, i.e., full-precision Resnet-18, BWN and XNOR-Net in [Rastegari et al., 2016], DoReFa-Net in [Zhou et al., 2016] and BNN in [Courbariaux et al., 2016] respectively. All comparative models use Resnet-18 as network topology. The full-precision Resnet-18 achieves 69.3% Top-1 accuracy. Although Rastegari et al. [2016]'s BWN model and DeReFa-Net perform well, it should be noted that they use full-precision and 4-bit activation respectively. Models (XNOR-Net and BNN) that used both binary weights and activations achieve much less satisfactory accuracy, and is significantly outperformed by ABC-Net with multiple binary weight bases and activations. It can be seen that ABC-Net has achieved state-of-the-art performance as a binary model.

One might argue that 5-bit width quantization scheme could reach similar accuracy as that of ABC-Net with 5 weight bases and 5 binary activations. However, the former one is less efficient and requires distinctly more hardware resource. More detailed discussions can be found in Section 5.2.

## 5 Discussion

### 5.1 Why adding a shift parameter works?

Intuitively, the multiple binarized weight bases/activations scheme works because it allows more information passing through. Consider the case that a real value, say 1.5, is passed to a binarized function $f(x) = \text{sign}(x)$, where sign maps a positive $x$ to 1 and otherwise -1. In that case, the outputs of $f(1.5)$ is 1, which suggests that the input value is positive. Now imagine that we have two binarization function $f_1(x) = \text{sign}(x)$ and $f_2(x) = \text{sign}(x - 2)$. In that case $f_1$ outputs 1 and $f_2$ outputs -1, which suggests that the input value is not only positive, but also must be smaller than 2. Clearly we see that each function contributes differently to represent the input and more information is gained from $f_2$ compared to the former case.

From another point of view, both coefficients ($\beta$'s) and shift parameters are expected to learn and utilize the statistical features of full-precision tensors, just like the scale and shift parameters in batch normalization. If we have more binarized weight bases/activations, the network has the capacity to approximate the full-precision one more precisely. Therefore, it can be deduced that when $M$ or $N$ is large enough, the network learns to tune itself so that the combination of $M$ weight bases or $N$ binarized activations can act like the full-precision one.

### 5.2 Advantage over the fixed-point quantization scheme

It should be noted that there are key differences between the multiple binarization scheme ($M$ binarized weight bases or $N$ binarized activations) proposed in this paper and the fixed-point quantization scheme in the previous works such as [Zhou et al., 2016, Hubara et al., 2016], though at first

---

Courbariaux et al. [2016] did not report their result on ImageNet. We implemented and presented the result.

thought $K$-bit width quantization seems to share the same memory requirement with $K$ binarizations. Specifically, our $K$ binarized weight bases/activations is preferable to the fixed K-bit width scheme for the following reasons:

(1) The $K$ binarization scheme preserves binarization for bitwise operations. One or several bitwise operations is known to be more efficient than a fixed-point multiplication, which is a major reason that BNN/XNOR-Net was proposed.

(2) A $K$-bit width multiplier consumes more resources than $K$ 1-bit multipliers in a digital chip: it requires more than $K$ bits to store and compute, otherwise it could easily overflow/underflow. For example, if a real number is quantized to a 2-bit number, a possible choice is in range {0,1,2,4}. In this 2-bit multiplication, when both numbers are 4, it outputs $4 \times 4 = 16$, which is not within the range. In [Zhou et al., 2016], the range of activations is constrained within [0,1], which seems to avoid this situation. However, fractional numbers do not solve this problem, severe precision deterioration will appear during the multiplication if there are no extra resources. The fact that the complexity of a multiplier is proportional to THE SQUARE of bit-widths can be found in literatures (e.g., sec 3.1.1. in [Grabbe et al., 2003]). In contrast, our $K$ binarization scheme does not have this issue – it always outputs within the range {-1,1}. The saved hardware resources can be further used for parallel computing.

(3) A binary activation can use spiking response for event-based computation and communication (consuming energy only when necessary) and therefore is energy-efficient [Esser et al., 2016]. This can be employed in our scheme, but not in the fixed $K$-bit width scheme. Also, we have mentioned the fact that $K$-bit width multiplier consumes more resources than $K$ 1-bit multipliers. It is noteworthy that these resources include power.

To sum up, $K$-bit multipliers are the most space and power-hungry components of the digital implementation of DNNs. Our scheme could bring great benefits to specialized DNN hardware.

## 5.3 Further computation reduction in run-time

On specialized hardware, the following operations in our scheme can be integrated with other operations in run-time and further reduce the computation requirement.

(1) Shift operations. The existence of shift parameters seem to require extra additions/subtractions (see (2) and (8)). However, the binarization operation with a shift parameter can be implemented as a comparator where the shift parameter is the number for comparison, e.g., $H_v(\boldsymbol{R}) = \begin{cases} \boldsymbol{1}, & \boldsymbol{R} \geq 0.5 - v; \\ -\boldsymbol{1}, & \boldsymbol{R} < 0.5 - v. \end{cases}$ ($0.5 - v$ is a constant), so no extra additions/subtractions are involved.

(2) Batch normalization. In run-time, a batch normalization is simply an affine function, say, $\mathrm{BN}(\boldsymbol{R}) = a\boldsymbol{R} + b$, whose scale and shift parameters $a, b$ are fixed and can be integrated with $v_n$'s. More specifically, a batch normalization can be integrated into a binarization operation as follow: $H_v(\mathrm{BN}(\boldsymbol{R})) = \begin{cases} \boldsymbol{1}, & a\boldsymbol{R} + b \geq 0.5 - v; \\ -\boldsymbol{1}, & a\boldsymbol{R} + b < 0.5 - v. \end{cases} = \begin{cases} \boldsymbol{1}, & \boldsymbol{R} \geq (0.5 - v - b)/a; \\ -\boldsymbol{1}, & \boldsymbol{R} < (0.5 - v - b)/a. \end{cases}$ Therefore, there will be no extra cost for the batch normalization.

## 6 Conclusion and future work

We have introduced a novel binarization scheme for weights and activations during forward and backward propagations called ABC-Net. We have shown that it is possible to train a binary CNN with ABC-Net on ImageNet and achieve accuracy close to its full-precision counterpart. The binarization scheme proposed in this work is parallelizable and hardware friendly, and the impact of such a method on specialized hardware implementations of CNNs could be major, by replacing most multiplications in convolution with bitwise operations. The potential to speed-up the test-time inference might be very useful for real-time embedding systems. Future work includes the extension of those results to other tasks such as object detection and other models such as RNN. Also, it would be interesting to investigate using FPGA/ASIC or other customized deep learning processor [Liu et al., 2016] to implement ABC-Net at run-time.

## 7 Acknowledgement

We acknowledge Mr Jingyang Xu for helpful discussions.

## Footnotes

* indicates corresponding author.

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
