[Supplementary Material]

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

# Supplementary Material

## S1 Summary of training algorithm in Section 3.3

---

**Algorithm 1** Training a $L$-layer ABC-Net. $c$ is the cost function for minibatch, and $\lambda$ is the learning rate decay factor. $\circ$ indicates element-wise multiplication. BatchNorm() specifies how to batch-normalize the output of convolution and BackBatchNorm specifies how to backpropagate through the normalization [Ioffe and Szegedy, 2015]. Analogously, Conv() and BackConv() respectively specify how to do convolution and how to backpropagate through the convolution. Update() specifies how to update the parameters when their gradients are known, such as ADAM [Kingma and Ba, 2014] and Momentum [Qian, 1999])

---

**Require:** a minibatch of inputs and targets, number of binary weight bases $M$, number of binary activations $N$, previous weights $\boldsymbol{W}$, and learning rate $\eta$.
**Ensure:** updated weights $\boldsymbol{W}$ and updated learning rate $\eta$.
  {1. Computing the parameters gradients:}
  {1.1. Forward propagation:}
  **for** $l = 1$ to $L$ **do**
    Compute $\alpha_m^l, \boldsymbol{B}_m^l, m = 1, \cdots, M$ with (2) and (3))
    $s^l \leftarrow \text{Conv}(\boldsymbol{W}^l, \boldsymbol{R}^{l-1})$ using (12)
    Optionally apply max-pooling
    $a^l \leftarrow \text{BatchNorm}(s^l)$
    **if** $l < L$ **then**
      $\boldsymbol{A}_n^l \leftarrow H_{v_n}(a^l), n = 1, 2, \cdots, N$ (using (11))
    **end if**
  **end for**
  {1.2. Backward propagation:}
  {Note that the gradients are full-precision.}
  Compute $g_{a_L} = \frac{\partial c}{\partial a_L}$
  **for** $l = L$ to $1$ **do**
    **if** $l < L$ **then**
      $g_{a^l} \leftarrow \sum_{n=1}^N \beta_n g_{\boldsymbol{A}_n^l} \circ \mathbb{I}_{0 \leq a^l - v_n \leq 1}$
    **end if**
    $g_{s_l} \leftarrow \text{BackBatchNorm}(g_{a^l}, s_l)$
    $g_{\beta_m}, g_{\boldsymbol{B}_m^l} \leftarrow \text{BackConv}(g_{s_l}, \boldsymbol{B}_m^{l-1}, \boldsymbol{A}_m^{l-1})$
  **end for**
  {2. Accumulating the parameters gradients:}
  **for** $l = 1$ to $L$ **do**
    With $g_{\boldsymbol{B}_m^l}$ known, compute $g_{\boldsymbol{W}^l}$ using (7)
    $\boldsymbol{W}^l \leftarrow \text{Update}(\boldsymbol{W}^l, \eta, g_{\boldsymbol{W}^l})$
    $\beta_m \leftarrow \text{Update}(\beta_m, \eta, g_{\beta_m}), m = 1, 2, \cdots, M$
    $v_m \leftarrow \text{Update}(v_m, \eta, g_{v_m}), m = 1, 2, \cdots, M$
    $\eta \leftarrow \lambda \eta$
  **end for**

---

## S2 Weight approximation

In this section we explore how well the weight approximation can achieved given adequate binary weight bases (Section 3.1). To gain a visual intuition, we randomly sample a slice of weight tensor from a full-precision Resnet-18 model pretrained on ImageNet. The sliced tensor is then vectorized, and we approximate it with $M$ bases using linear regression (see (3)). The results are presented in Figure S2, where the left subfigure shows the root mean square (RMSE) for the estimated weights with increasing number of bases, and the right one shows 5 fitting results, whose choice of $M$ are respectively 1 to 5 from top to bottom. The blue line in the right subfigure draws the groundtruth weights from the full-precision pretrained model, and the red line is the estimated one. It can be observed that $M = 3$ is adequate to have a rough fitting, and it gets almost perfect when $M = 5$.

Figure S2: Fitting a section of weights of full-precision Resnet-18 trained on ImageNet. On the left side the RMSE for the estimated weights with increasing number of bases is shown, and on the right side 5 fitting results are shown, whose choice of $M$ are respectively 1 to 5 from top to bottom (Blue line: full-precision weights; Red line: estimated weights).

## S3 Feature map

It is also possible to perform more complex tasks beyond classification using ABC-Net, as long as the model is built upon a CNN, such as faster RCNN for object detection, in which the classification model serves as a pre-train model. Thus, one might be interested in whether ABC-Net learns similar feature maps as its full-precision counterpart. Figure S3 shows several example image and the corresponding feature maps of these two models, from which we see that they are indeed similar. This shows the potential for ABC-Net to further generalize on more complex tasks mentioned above.

## S4 Parameter settings for the experiment in Section 4.1.2

The parameters $u_i$'s, the initial values for $\beta_n$'s and $v_n$'s can be treated as hyperparameters. At the beginning of our exploration we randomly choose these initial values. Bit by bit we began to find certain patterns to achieve good performance: for $u_i$'s, usually symmetric; for $v_n$'s, maybe slightly shift towards the negative direction. These are based on tunings and also the observation of the full-precision distribution of weights/activations. Table S4 provides the parameter settings for the experiment in Section 4.1.2. All ABC-Net models in the experiments are trained using SGD with momentum, and the initial learning rate is set to 0.01.

## S5 Relationship between accuracy and number of binary weight bases $M$

Figure S4 shows that the relationship between accuracy and the number of binary weight bases $M$ appears to be linear. Note that we keep the activations being full-precision in this experiment.

## S6 Application on visual perception of forest trails

### S6.1 VGG-like Network Topology

A VGG-like network topology is used for visual perception of forest trails as illustrated in Figure S5.

### S6.2 Experiment results on visual perception of forest trails dataset

Giusti et al. [2016] cast the forest or mountain trails perception problem for mobile robots as a image classification task based on Deep Neural Networks. The dataset is composed by 8 hours of $1920 \times 1080$ 30fps video acquired by a hiker equipped with three head-mounted cameras . Each

Figure S3: Examples of feature maps. The feature maps from the first convolution layer of ABC-Net (above) looks similar to that of its full-precision counterpart (below). Settings for the ABC-Net: $M = 5, N = 3$, using Resnet-18.

Table S4: Parameter settings for the experiment in Section 4.1.2. "res18", "res34" and "res50" are short for Resnet-18, Resnet-34 and Resnet-50 network topology respectively. $M$ and $N$ refer to the number of weight bases and activations respectively.

| Network | $M$ | $N$ | shift parameters ($u_i$'s) | shift parameters ($v_i$'s) | $\beta$'s |
|---------|-----|-----|----------------------------|----------------------------|-----------|
| res18 | 1 | 1 | 0 | 0.0 | 1.0 |
| res18 | 3 | 1 | -1,0,1 | 0.0 | 1.0 |
| res18 | 3 | 3 | -1,0,1 | -1.5, 0.0, 1.5 | 1.0, 1.0, 1.0 |
| res18 | 3 | 5 | -1,0,1 | -3.5, -2.5, -1.5, 0.0, 2.5 | 1.0, 1.0, 1.0, 1.0, 1.0 |
| res18 | 5 | 1 | -2,-1,0,1,2 | 0.0 | 1.0 |
| res18 | 5 | 3 | -2,-1,0,1,2 | -0.9, 0.0 0.9 | 1.0,1.0,1.0 |
| res18 | 5 | 5 | -1,-0.5,0,0.5,1 | -3.5, -2.5, -1.5, 0.0, 2.5 | 1.0, 1.0, 1.0, 1.0, 1.0 |
| res34 | 3 | 3 | -1,0,1 | -3.0, 0.0, 3.0 | 1.0, 1.0, 1.0 |
| res34 | 5 | 5 | -1,-0.5,0,0.5,1 | -3.5, -2.5, -1.5, 0.0, 2.5 | 1.0, 1.0, 1.0, 1.0, 1.0 |
| res50 | 5 | 5 | -1,-0.5,0,0.5,1 | -3.5, -2.5, -1.5, 0.0, 2.5 | 1.0, 1.0, 1.0, 1.0, 1.0 |

image is labelled in one of three classes: turn right, go straight, turn left. We evaluate ABC-Net against its full precision counterpart using this dataset. The classification result is shown in Table S5 by fixing both number of weight bases $M$ and activation bases $N$ to be 5.

Figure S4: Top-1 (left) and Top-5 (right) accuracy of ABC-Net on ImageNet, using full-precision activation and different choices of the number of binary weight bases $M$.

Table S5: Classification accuracy on Forest Trails dataset. 'FP' stands for 'Full Precision'.

| Network | shift parameters ($u_i$'s) | shift parameters ($v_i$'s) | $\beta$'s | ABC-Net | FP |
|---------|---------------------------|---------------------------|-----------|---------|------|
| VGG-like | -1,0.5,0.0,0.5,1 | 0,0,0,0,0 | 1,1,1,1,1 | 78.0% | 77.7% |

BatchNorm
↓
Convolution, filter:$4 \times 4$, output channel: 64
↓
MaxPooling, filter:$2 \times 2$, strike: $2 \times 2$
↓
BatchNorm
↓
ReLu
↓
Convolution, filter:$4 \times 4$, output channel: 64
↓
MaxPooling, filter:$2 \times 2$, strike: $2 \times 2$
↓
BatchNorm
↓
ReLu
↓
Convolution, filter:$4 \times 4$, output channel: 128
↓
MaxPooling, filter:$2 \times 2$, strike: $2 \times 2$
↓
BatchNorm
↓
ReLu
↓
Convolution, filter:$4 \times 4$, output channel: 128
↓
MaxPooling, filter:$2 \times 2$, strike: $2 \times 2$
↓
BatchNorm
↓
ReLu
↓
Convolution, filter:$4 \times 4$, output channel: 128
↓
MaxPooling, filter:$2 \times 2$, strike: $2 \times 2$ BatchNorm
↓
ReLu
↓
Convolution, filter:$4 \times 4$, output channel: 128
↓
MaxPooling, filter:$2 \times 2$, strike: $2 \times 2$
↓
BatchNorm
↓
ReLu
↓
FullyConnected, output channel: 200
↓
Dropout, ratio: 0.5
↓
BatchNorm
↓
ReLu
↓
FullyConnected, output channel: 3

Figure S5: The network topology for visual perception of forest trails.