[Reviews · NeurIPS 2017]

Reviewer 1



The paper describes a new method for training binary convolutional neural networks. The scheme addresses both using binary weights and activations. The key insight here is to approximate the real valued weights via a linear combination of M binary basis weights. The coefficients for reconstructing the real weights can be found using least squares in the forward pass, and then pulled outside the convolution to allow for fast binary convolution at test time. A similar approach is taken for the activations, but in this case the weights and shifts are trained as normal during backpropagation. The result is a network that requires M more binary convolutions than a straightforward binary neural network, but it is expected that these will be significantly more hardware friendly. Experiments on ImageNet show that the approach outperforms competing approaches. Binary convolutional networks are an important topic with with obvious commercial applications. The idea in the paper is good, and the results on ImageNet are encouraging. Comments 1. Given that you approximate a real valued convolution with M binary convolutions, it seems to me that the approximate cost would be similar to using M binary CNNs like XNOR nets. With this in mind, I think the approach should really be compared to an ensemble of M XNOR nets trained with different initial weights, both in terms of total train time and prediction accuracy. 2. The paper suggests the v_i shifts are trainable. Why are values for these given in table 4 and the supplementary material. Are these the initial values? 3. The paper uses fixed values for the u_i's but mentions that these could also be trained. Why not train them too? Have you tried this? 4. It's not clear from the paper if solving regression problems for the alphas during training adds much to the train time. 5. It would be nice to have an estimation of the total cost of inference for the approach on a typical net (e.g. ResNet-18) given some realistic assumptions about the hardware, and compare this with other methods (perhaps in the supplementary material)

Reviewer 2



This paper proposes a scheme to approximate the weights of a neural network layer by the linear combination of multiple layers with binary {1, -1} weights. The proposed solution affects the inference time performance of the network only, but it relies on batch-normalization at training time to improve its binarizability. The solution makes use of both local optimization of the weight matrices as well as using backpropagation to optimize them in their global context as well. A nice property of the approach is that even it can result on a speedup even on current hardware, but it has very good potential for even greater speed and power gains with custom hardware as it relies on very cheap and well parallelizable operations. Given the fact that this is the first solution that can give comparable recognition performance while massively cutting the raw computational cost of the network (using 5 binary operations for each floating point), and its great potential when combined with custom hardware, this paper could en up having a great impact on the design of custom accelerators and mobile vision in general. The only weakness is that the quality was tested against a relatively weak baseline: their full precision reference model reaches ~69% top-1 accuracy on the ILSVRC2012 dataset while the current state of the art is over 80% top-1 accuracy.

Reviewer 3



This paper extends previous work done for binary CNNs using multiple binary weight/activation matrices. Although it is presents incremental additions to prior work, the paper shows strong large-scale results on the standard ImageNet benchmark as well as thorough experimentations, and a clear presentation. This paper would entertain a wide interest among NIPS audience. Comments and suggestions for improvements: - You stated that the B and v parameters of activations are learned by the NN, but in table S5, the learned Bs are all 1.0 except for one row, while v values are almost symmetric. Could you elaborate further on this point, and update the text to state the exact procedure you use for learning/setting these parameters. - Could you provide a histogram of the discovered alpha values for weight matrices, to gain more insight on how sparse they are? - How bad are the results if you don’t initialize by the FP CNN? Do you need to start from a fully converged model, or just couple epochs are sufficient? - Given the strong correlation between the number of binary weight matrices M and final performance (as shown in Table 1, and figure S4), Could you provide further experimental results to show the performance drop when M > 5 (as stated in section 3.3) and the potential effect of regularization to fix it? - In section S1, algorithm 1: you need to add gradient and update for the v parameters (based on the text you learn them by backpropagation). - In section S5, please state if you fix the activations to FP.